# Monitoring of Water Quality, Antibiotic Residues, and Antibiotic-Resistant *Escherichia coli* in the Kshipra River in India over a 3-Year Period

**DOI:** 10.3390/ijerph17217706

**Published:** 2020-10-22

**Authors:** Nada Hanna, Manju Purohit, Vishal Diwan, Salesh P. Chandran, Emilia Riggi, Vivek Parashar, Ashok J. Tamhankar, Cecilia Stålsby Lundborg

**Affiliations:** 1Department of Global Public Health, Health Systems and Policy (HSP): Medicines Focusing Antibiotics, Karolinska Institutet, 171 77 Stockholm, Sweden; vishal.diwan@ki.se (V.D.); ejetee@gmail.com (A.J.T.); Cecilia.Stalsby.Lundborg@ki.se (C.S.L.); 2Department of Pathology, R.D. Gardi Medical College, Ujjain 456006, India; 3Department of Public Health and Environment, R.D. Gardi Medical College, Ujjain 456006, India; vivek.p@rdgmc.edu.in; 4ICMR—National Institute for Research in Environmental Health, Bhopal 462030, India; 5HLL Lifecare Ltd., Kharghar Navi, Mumbai 410210, India; saleshp@gmail.com; 6SSD Epidemiologia screening—CPO, University Hospital ‘Cittàdella Salute della Scienza’, 10126 Turin, Italy; emilia.riggi@cpo.it; 7Indian Initiative for Management of Antibiotic Resistance, Department of Environmental Medicine, R.D. Gardi Medical College, Ujjain 456006, India

**Keywords:** antibiotic residues, antibiotic-resistant *E. coli*, water quality, river water and sediment, environmental pollution

## Abstract

The emergence of antibiotic resistance is a major global and environmental health issue, yet the presence of antibiotic residues and resistance in the water and sediment of a river subjected to excessive anthropogenic activities and their relationship with water quality of the river are not well studied. The objectives of the present study were a) to investigate the occurrence of antibiotic residues and antibiotic-resistant *Escherichia coli* (*E. coli*) in the water and sediment of the Kshipra river in India at seven selected sites during different seasons of the years 2014, 2015, and 2016 and b) to investigate the association between antibiotic residues and antibiotic-resistant *E. coli* in water and sediment and measured water quality parameters of the river. Antibiotic residues and resistant *E. coli* were present in the water and sediment and were associated with the measured water quality parameters. Sulfamethoxazole was the most frequently detected antibiotic in water at the highest concentration of 4.66 µg/L and was positively correlated with the water quality parameters. Significant (*p* < 0.05) seasonal and spatial variations of antibiotic-resistant *E. coli* in water and sediment were found. The resistance of *E. coli* to antibiotics (e.g., sulfamethiazole, norfloxacin, ciprofloxacine, cefotaxime, co-trimoxazole, ceftazidime, meropenem, ampicillin, amikacin, metronidazole, tetracycline, and tigecycline) had varying associations with the measured water and sediment quality parameters. Based on the results of this study, it is suggested that regular monitoring and surveillance of water quality, including antibiotic residues and antibiotic resistance, of all rivers should be taken up as a key priority, in national and Global Action Plans as these can have implications for the buildup of antibiotic resistance.

## 1. Introduction

Antibiotic residues have been recognized in recent years as important emerging environmental contaminants because of their potential adverse ecological and human health effects through the development of antibiotic resistance [1,2,3]. The exposure of bacteria to antibiotic residues in the environment contributes to the selection of resistance, and the environment acts as a reservoir and a potential transmission route for resistant bacteria [4,5]. Various factors affect the concentrations of detected antibiotics in the aquatic environment, such as the use of antibiotics in different periods over time and environmental behaviors of antibiotics such as degradation and sorption [6,7].

Antibiotic residues and antibiotic-resistant bacteria enter the aquatic environment, including river water [8], through various pathways such as discharge of industrial effluent [9,10,11], hospital and municipal wastewater [12,13], and agricultural runoff [14]. Rivers appear to be a reservoir of antibiotic resistance and play an important role in transportation of antibiotic resistance between various environmental compartments [15,16]. River water might create possible pathways for antibiotic resistance transmission between the environment, humans, and animals [17]. In this regard, similar patterns in antibiotic resistance of *Escherichia coli* (*E. coli*) isolates from humans, animals, and their water environment have been reported [18,19]. *E. coli* is a useful indicator of fecal contamination and is considered a reservoir of antibiotic resistance in bacterial communities [20].

Kshipra river is the major source of water supply for the Ujjain district in India, where the water is used for drinking and irrigation purposes [21]. Kshipra river is one of the holy Indian rivers which hosts various mass bathing events including Simhastha Kumbh Mela, the largest mass-gathering event which is organized every twelve years in Ujjain city. Pilgrims from all over the world come to bathe in this holy river and perform various activities including worship rituals. As a consequence, the water quality of the Kshipra river remains depleted [22,23]. We have previously reported seasonal variation in antibiotic residues found in the Kshipra river over a one-year period [24]. However, knowledge of the occurrence of antibiotic residues and resistance in water and sediments and their association with the water quality parameters of the Kshipra river in the longer-term is lacking.

The objective of the present study was to investigate and monitor the occurrence of antibiotic residues and antibiotic-resistant *E. coli* in water and sediments and the water quality of the Kshipra river in central India during various seasons and at various sites over a 3-year period.

## 2. Materials and Methods 

The methods are described in brief here; a detailed description of the methods is available in the published protocol [25].

### 2.1. Setting

The study was conducted in Ujjain district of the Indian state of Madhya Pradesh. Madhya Pradesh is the second largest state in India with almost 77% of its population of 72 million living in rural areas with a low Human Development Index (HDI) score of 0.45 [26]. 

Specifically, the study was conducted within the reaches of the Kshipra river that flows through the city of Ujjain. The reach extends from Triveni Ghat to Kaliyadeh Palace, the respective entry and exit points of the river for the city of Ujjain. 

A tributary of Kshipra, the Khan river, pollutes the water of Kshipra by industrial pollutants. In order to keep Kshipra water clean from river Khan for the purposes of mass bathing, especially during Simhastha-Kumbh Mela, which was held in Ujjain city from 22 April to 21 May 2016, the Khan river was diverted through a pipeline in mid-March 2016 and water of another river—Narmada (which carries cleaner water)—was diverted to Kshipra river in Ujjain from 25 February to 21 May 2014 and from 17 April to 21 May 2016 [27].

### 2.2. Sample Collection

Water samples from the river were collected in duplicate in sterile containers from seven selected sites. Using an Ekman Dredge sediment sampler, sediment samples were also collected from the same seven sites (Figure 1, Appendix A). In total, 3.2 L of water was collected from each sampling point: 1 L for the analysis of antibiotic residues, 2 L for other water-quality estimations, and 200 mL for colony counts and antibacterial-susceptibility tests. In addition, 2 kg of sediment was collected from each sampling point: 1.5 kg for the analysis of antibiotic residues, 450 g for the sediment quality parameters, and 50 g for colony counts and antibacterial-susceptibility tests [28]. The criteria for the sampling sites included both point and non-point sources of pollution, and the locations were chosen at places which have industries and agriculture activities, as well as at the confluence of the Khan and Kshipra rivers. Khan river brings pollutants from pharma factories nearby. Further, there were mass-bathing spots or spiritually important places where the pollution load was expected to be high due to the bathing of people. Sampling was conducted once during each of the four seasons for three consecutive years: summer (29 May 2014, 25 May 2015, 26 May 2016), rainy (15 July 2014, 27 July 2015, 1 August 2016), autumn (10 October 2014, 10 October 2015, 17 October 2016), and winter (22 December 2014, 29 December 2015, 2 January 2017). 

### 2.3. Transport of Water and Sediment Samples to the Laboratory

Samples for antibiotic residues were transported to the laboratories of the Shriram Institute for Industrial Research in New Delhi within 12 h of sample collection in screw-capped amber bottles wrapped in silver foil. Samples for studying the water-quality parameters and antibiotic susceptibility were collected in plastic cans and sterilized 200 mL glass bottles, respectively. Sediment samples for colony counts and antibiotic susceptibility and for other parameters were collected in 50-g sterilized conical-bottomed tubes and sterile 1 L plastic containers, respectively [29].

### 2.4. Water Quality Parameters Measured in the Field

The following water quality parameters were measured immediately after the collection of samples: pH, total dissolved solids (TDS), conductivity (Cond), free carbon dioxide (Free CO_2_), ambient temperature (Abtemt), water temperature (Wtemp), and dissolved oxygen (DO). The ambient and water temperatures were measured using a mercury (Hg) thermometer graduated to 100 °C with an accuracy of 0.1 °C–0.2 °C. The pH, conductivity and TDS was measured using an ECO Tester hand-held digital pH meter (Thermo Fisher Scientific, Mumbai, India), a 611-El Digital Conductivity Meter (Electronic India, Parwanoo, India), and a digital TDS meter (AM-TDS-01, Aquasol Digital, Rakiro Biotech Systems PVT LTD, Navi Mumbai, India), respectively. The titration method was used to measure DO, free CO_2_, and carbonate alkalinity as per standard methods [25,30]. 

### 2.5. Water and Sediment Quality Parameters Examined in the Laboratory

Total hardness (TH), chloride (Cl), turbidity (Turb), nitrate nitrogen (NO_3_-N), total alkalinity (Talk), methyl orange alkalinity (HCO_3_ Alk), total phosphorus (TP), ortho phosphorus (OrthoP), organic phosphorus (OrgP), available phosphorus (AP), chemical oxygen demand (COD), biochemical oxygen demand (BOD), total suspended solids (TSS), phenolphthalein alkalinity (CO_3_Alk), calcium hardness (CaH), magnesium hardness (MgH), organic matter (OM), soluble bicarbonate (SolCO_3_), total coliform (TCC), and total *E. coli* (TEC) were examined at the Central Research Laboratory of R.D. Gardi Medical College, Ujjain [31,32].

A titration method was used to measure TH, Cl, and CaH. TSS was measured by filtering a known quantity of sampled water through a pre-weighed filter and then weighing the filter paper after drying at 105 °C. The 5-d BOD at 20 °C was measured. The open reflex method was applied to measure COD, while NO_3_-N, TP, and the available phosphorus were determined by a spectrophotometer (UV-1800 Shimadzu, Kyoto, Japan). Turbidity was determined by the Nephelo Turbidity Meter (Deluxe turbidity meter-385, Electronic India, Prwanoo, India). All sediment analysis of the various parameters was carried out in the laboratory. The collected sediment samples were first air-dried at room temperature before any analysis. Sediment samples were analysed for pH, Cl, SolCO_3_, NO_3_-N, AP, and OM.

### 2.6. Antibiotic Residue Analysis

River water and sediment samples were analyzed for the presence of antibiotics: ceftriaxone, ofloxacin, norfloxacin, ciprofloxacin, sulfamethoxazole, metronidazole, and total residual antibiotics as β-lactam. These antibiotics were selected based on antibiotic residues previously found in the same geographical area [29,33], environmental stability, and known and suspected environmental impacts of the antibiotic and the degree of antibiotic metabolism [34]. 

In brief, every water sample was homogenized by mixing thoroughly. The homogenized sample (50 mL) was filtered through a 0.45 µm membrane filter paper. The sample was acidified with 1 N H_2_SO_4_ to pH 3 and then loaded on an activated C-18 cartridge (activated with 5 mL methanol, 5 mL methanol/water (50:50) followed by 5 mL of acidified water at pH 3). The cartridge was washed with 5 mL of acidified water, and adsorbed compounds were eluted with 5 mL of 5% trimethylamine in methanol. The eluent was evaporated to dryness with a gentle stream of nitrogen gas at 50 °C. The residue was reconstituted with acetonitrile to make final volume of 1 mL.

Sediment samples were homogenized by mixing thoroughly. Acidified water (100 mL, pH 3.5 adjusted with phosphoric acid) was added to 20 g of samples shaken for 1 h and filtered with the help of filtration assembly through Whatman filter paper no. 41. The sample was loaded on an activated C-18 cartridge. The cartridge was washed with 5 mL of acidified water, and adsorbed compounds were eluted with 5 mL of 5% triethylamine in methanol. The eluent was evaporated to dryness with a gentle stream of nitrogen gas at 50 °C. The residue was reconstituted with acetonitrile to 2 mL.

Antibiotic residues were detected by using solid-phase extraction followed by liquid chromatography tandem-mass spectrometry (LC-MS/MS) (Waters 2695 Series Alliance Quaternary Liquid Chromatography System, Waters, Milford, MA, USA) with a triple quadruple mass spectrometer (Quatro-micro API, Micromass, Manchester, UK) equipped with an electro-spray interface and Masslynx 4.1 software (Micromass, Manchester, UK) for data acquisition and processing. The respective limits of quantification (LOQ in µg/L) and limits of detection (LOD µg/L) for antibiotics tested in water samples were as follows: metronidazole—0.05 and 0.01, sulfamethoxazole—0.08 and 0.01, norfloxacin—0.1 and 0.01, ciprofloxacin—0.1 and 0.01, ofloxacin—0.1 and 0.01, ceftriaxone—20 and 1. The respective limits of quantification (LOQ µg/kg) and limits of detection (LOD ug/kg) for antibiotics tested in sediment samples were as follows: metronidazole—0.05 and 0.05, sulfamethoxazole—0.08 and 0.05, norfloxacin—0.1 and 0.05, ciprofloxacin—0.1 and 0.05, ofloxacin—0.1 and 0.05, ceftriaxone—20 and 0.25.

### 2.7. Microbiological Methods

The received samples were processed as follows: (1) Ten-fold serial dilutions (1:100, 1:1000 as per turbidity of samples) of surface water in 0.9% sterile saline (NaCl) solution. The whole sediment sample was added to 100 mL of 0.9% normal saline, and then serial ten-fold dilutions were carried out. The diluted samples were filtered following a standard membrane-filtration technique using nylon membrane filters of 47 mm in diameter with a pore size 0.45 µm. After filtration, the membrane was taken out from the assembly and placed on selective and differential media for the identification and isolation of *E. coli* and non-*E. coli* isolates using HiCrome coliform agar with incubation at 37 °C for 24 h. (2) Bacterial enumeration was carried out to estimate the total coliform count and total *E. coli* count in colony-forming units (CFUs) per 100 mL on agar. (3) The isolation and subsequent DNA extraction of six *E. coli* isolates per surface water and sediment sample were used for polymerase chain reaction (PCR) testing. (4) Susceptibility tests for eight different classes of antibiotics inclusive of ampicillin, cefotaxime, ceftazidime, cefepime, nalidixic acid, ciprofloxacin, nitrofurantoin, gentamicin, amikacin, tetracycline, tigecycline, imipenem, meropenem, co-trimoxazole, and sulfamethiazole were conducted using the Kirby Bauer disc-diffusion test on Muller Hinton (MH) agar by using a bacterial suspension with 0.5 McFarland turbidity [35]. Clinical and Laboratory Standard Institute (CLSI) guidelines were used to measure and interpret the zone diameter of bacterial growth inhibition [36]. *E. coli* ATCC25922 strain was used as a control when testing AST for *E. coli and* an extended spectrum beta-lactamases (ESBL)-producing *K. pneumoniae* ATCC 700603 strain as per CLSI guidelines. The diameter of inhibition zones according to manufacturer’s details (HiMedia Laboratories Pvt. Ltd., Mumbai, India) was measured to the nearest millimetre by two technical experts independently with quality measures as described in detail previously [25]. 

### 2.8. Molecular Methods

The DNA extraction of all *E. coli* isolates was carried out using the heat lysis method for the detection of antibiotic resistance coding genes. Phenotypically classified *E. coli* was tested for the presence of various genes such as ESBL-coding (*blaCTX-M, blaSHV,* and *blaTEM*), plasmid-mediated quinolone resistance (*qnrA, qnrB,* and *qnrS*), carabapenemase resistance (VIM and NDM), and sulfonamide resistance genes (sul I and sul II); the primer details and PCR and phylogenetic grouping (*chuA, yjaA,* and *TspE4C2*) were presented in detail previously [24,37,38,39] and are briefly shown in Appendix A.

### 2.9. Data Management and Statistical Analysis

Descriptive statistics are used to present data as means and the range for continuous data, while listed categorical variables are presented as numbers and percentages. Analysis of variance (ANOVA) was conducted to determine seasonal variation in antibiotic residues and resistance among the seven sites over a 3-year period. A post-hoc analysis was applied to test the difference in antibiotic residues and resistance between seasons within each year. Tukey correction was used to adjust *p*-values for multiple pairwise comparison. Correlations between antibiotics and water quality parameters were analyzed with Pearson’s rank correlation test. The results are presented in tables with corresponding *p*-values, and significant associations were determined by *p*-values < 0.05. All analyses were performed in R 3.4.1 (R Foundation for Statistical Computing, Vienna, Austria) [40].

The study was approved by the Ethics Committee of the R.D. Gardi Medical College, Ujjain, MP, India (No: 2013/07/17-311). Water and sediment samples were taken from public places and no specific permission was required for this purpose.

## 3. Results

### 3.1. Antibiotic Residues in River Water and Sediments

The numbers of samples with antibiotic residues detected in different seasons in water and sediment are presented in Appendix A. The mean concentrations of the antibiotics detected in the water of Kshipra river in various seasons from the seven sampling sites over a 3-year period are presented in Figure 2 and Appendix A.

In water, norfloxacin was detected at the highest levels in autumn (0.98 µg/L) at site 2 of the first year, and ofloxacin was detected at the highest levels in autumn (1.46 µg/L) at site 5 of the first year compared to all other samples. The concentrations of metronidazole were relatively low as compared to other antibiotics, with the highest level of 0.27 µg/L detected in the rainy season at site 6 of the second year. Sulfamethoxazole was detected most consistently in different seasons and sites of the first and second year, with the highest concentration of 4.66 µg/L found in autumn at site 2 of the first year. Samples showed total residual antibiotics as β-lactam (>5 ppb) in summer of the first year at site 1, 2, 3, 4, 5, and 7. In addition, the concentration levels of sulfamethoxazole and ofloxacin were significantly associated with different seasons, sites, and years (*p* < 0.05). In sediment, the only antibiotics that were detected in the samples were ofloxacin (9.74 µg/Kg) in winter at site 4 of the first year and sulfamethoxazole (8.23 µg/Kg) in winter at site 3 of the first year. In addition, samples showed total residual antibiotics as β-lactam (>5 ppb) in the rainy season of the first year at site 1, 3, 5, and 7.

Seasonal pairwise comparisons of antibiotic residue concentrations in water are presented in Appendix A. Significant seasonal pairwise differences in sulfamethoxazole concentrations of the first and second year (*p* < 0.05) were found. 

### 3.2. Antibiotic-Resistant E. coli and Resistance Genes in River Water and Sediments

Antibiotic-resistant *E. coli* was present in the water and sediment during various seasons and at various sampling sites over the 3-year period. In water, significant (*p* < 0.05) seasonal and spatial variations in the resistance of *E. coli* to ampicillin, cefepime, amikacin, tetracycline, meropenem, nalidixic acid, co-trimoxazole, and sulfamethizole were found (Figure 3). There were significant differences in the occurrence of multidrug-resistant (MDR) *E. coli* during various seasons (Table 1).

In *E. coli* from sediment samples, significant (*p* < 0.05) seasonal and spatial variation in the resistance was found to ampicillin, cefotaxime, ceftazidime, meropenem, and nitrofurantoin (Figure 4). There were significant differences in the occurrence of extended spectrum β-lactamase (ESBL) and MDR *E. coli* during various seasons (Appendix A). 

In the water and sediment, significant seasonal pairwise differences in antibiotic-resistant *E. coli* were observed over the three years (Appendix A). 

Antibiotic resistance genes were analyzed from river water and sediment samples. The ESBL-coding gene, *blaCTX-M1*, was detected in different seasons from 19–66% of cephalosporin-resistant *E. coli* isolates from water and 13–60% isolates from sediment. *blaCTX-M9* was found in water in rainy and summer seasons of the second and third year, respectively.

The *qnrS* plasmid-mediated quinolone-resistance gene was the most commonly detected gene. The *qnr A* and carbapenems coding genes *VIM* and *NDM* were not detected in any sample while *sul-I* and *sul-II* genes were present in 5–31% isolates from river water and 20–80% isolates from river sediment. Most *E. coli* isolates from both river water and sediment belonged to phylogenetic groups A and B1 (Table 2, Table 3 and Table 4).

### 3.3. Correlation between Antibiotic Residues and Water Quality and Correlation between Antibiotic-Resistant E. coli and Water Quality

The means and standard deviations of the water quality parameters observed in different seasons and sites in the case of water of the Kshipra river over a 3-year period are presented in Table 5 and Appendix A. Significant (*p* < 0.05) seasonal and spatial variations in water quality parameters were observed.

Pearson’s rank correlation test for relationships between water quality parameters and antibiotic residues indicated that sulfamethoxazole was significantly correlated with the water quality parameters (*p* < 0.05), except for free carbon dioxide (free CO_2_) and phenolphthalein alkalinity (CO_3_alk). Ofloxacin residues were also significantly associated with some measured water quality parameters (Appendix A).

In addition, correlation tests were performed for measured water and sediment quality parameters and antibiotic-resistant *E. coli*. The resistance of *E. coli* to antibiotics (e.g., sulfamethiazole, norfloxacin, ciprofloxacine, cefotaxime, co-trimoxazole, ceftazidime, meropenem, ampicillin, amikacin, metronidazole, tetracycline, and tigecycline) had varying associations with measured water and sediment quality parameters such as water temperature, air temperature, pH, conductivity, dissolved oxygen, total dissolved solids, turbidity, phenolphthalein alkalinity, chloride, total hardness, ortho phosphorus, and soluble bicarbonate (Appendix A).

## 4. Discussion

To our knowledge, this is the first study that has followed antibiotic residues, antibiotic resistance in *E. coli,* and water quality over a three-year period in water and sediment of one of the holy rivers of India, Kshipra river. Antibiotic residues were present in both the water and sediment of the Kshipra river over the 3-year period. Significant seasonal and spatial variation in antibiotic residues was found in water. The water quality parameters had varying associations with antibiotic residues and antibiotic-resistant *E. coli*. 

The occurrence and levels of antibiotic residues in the water and sediment of Kshipra river varied by season and site over the 3-year period. Among the analyzed antibiotics, sulfamethoxazole, ofloxacin, norfloxacin, metronidazole, and total residual antibiotics as β- lactam were detected in water samples. In general, the qualitative and quantitative variation in the occurrence of antibiotics in river water and sediment can be ascribed to the variation of antibiotic consumption and anthropogenic activities in the capture area, the physiochemical behavior of the antibiotics in the aquatic environment, and agricultural, domestic and industrial activities along the river and its tributaries. For example, site 2 (Khan) is dominated by agricultural activities and is located on the banks of river Khan, a sub-tributary of Kshipra river. Khan is the major contributor of the contamination and degrading of the water quality of the Kshipra river [41]. Domestic as well as industrial waste water is disposed of without treatment into Khan river from the nearby Indore city every day, which in turn pollutes the Kshipra river [41]. In addition, several mass bathing events, including Simhastha Kumbh Mela, are organized on the banks of Kshipra river. Mass bathing occasions have the potential to deteriorate the river water quality, as millions of pilgrims take a bath in this holy river, leading to increased pollutants in the water [22]. Sulfamethoxazole was the most abundant antibiotic in water in different seasons and sites with the highest level in autumn at site 2 (Khan). Sulfamethoxazole is hydrophilic; it is characterized by high water solubility and is found to be the most persistent sulfonamide in water, with low biodegradability and high transport propensity [42,43]. Sulfamethoxazole, due to its low adsorption, can also be transported from soil or manure through surface runoff into river water [8]. A previous study also reported that the sulfonamides were the most detected antibiotic residues in the water of the Haihe river basin, China [8]. Fluoroquinolones, including norfloxacin and ofloxacin, were detected at higher levels in autumn at sites 2 (Khan) and 5 (Mangalnath), respectively. Site 5 is located near the ancient Mangalnath temple situated on the bank of the river. Anthropogenic activities such as bathing and washing by pilgrims are common features of this site. In the upstream zone of this sampling site, two or three small tributaries join river Kshipra and bring domestic and industry waste water of Ujjain city [44]. These could be contributory factors for the occurrence of fluoroquinolones. Fluoroquinolones show high sorption potential, poor water solubility, and very low biodegradability [45,46]. A previous study reported that up to 4.7 µg/L of norfloxacin and 10 µg/L of ofloxacin were found in water samples in the Isakavagu-Nakkavagu river, closed to pharmaceutical production plants in Hyderabad, India [47].

The occurrence of antibiotic residues in the environments is likely to lead to the development of antibiotic-resistant bacteria and resistance genes, which eventually can pose risks to human health [48]. In the present study, antibiotic-resistant *E. coli* was present in water and sediment in various seasons, sampling sites, and years. Significant seasonal and spatial variations of antibiotic-resistant *E. coli* in water and sediment were observed. The resistance of *E. coli* to antibiotics had varying associations with measured water and sediment quality parameters. *E. coli* has been proposed for monitoring of antibiotic resistance in aquatic environments [49]. The resistance of *E. coli*, including resistance genes, to different types of antibiotics differed in the water and sediment. This variation might be attributed to the different behaviours of antibiotics between water and sediments that are affected by the structures of the antibiotics and the physicochemical properties of the sediment. Sediment is a major sink and reservoir for antibiotics in water [50,51]. The profiles and the distribution of antibiotic-resistant bacteria and resistance genes are in general influenced by a combination of factors including the use and type of antibiotics, environmental behaviors of antibiotics, water quality, bacterial community structure, and anthropogenic activity [19,52,53,54]. The environmental release of antibiotics in effluents from various pollution sources, combined with direct contact between natural bacterial communities and discharged antibiotic-resistant bacteria and their resistance genes, is a driver for the emergence of resistant strains in the environment including rivers [55,56,57]. Antibiotics in the river water can act as selective pressure and a major driving force in aquatic environments to develop and maintain antibiotic-resistant bacteria and resistance genes. For example, the presence of sulfamethoxazole in the water of Kshipra river might potentially lead to the development of resistance to co-trimoxazole and sulfamethiazole in *E. coli* and the presence of sulfonamide resistance genes (*sul-I and sul-II*). The spread of antibiotic resistance genes along with ESBL-producing *E. coli* isolates into rivers is worrisome and contributes to the growing concerns about resistant bacteria and their potential risks to the environment and public health [58,59,60,61,62,63,64]. CTX-M-producing *E. coli* is the most predominant type of ESBL-producing *E. coli* from clinical and aquatic environments in recent years [65]. In the present study, the ESBL-coding gene (*blaCTX-M-1*) and *qnrS* plasmid-mediated quinolone-resistance gene were found as the most the prevalent in river water and sediment. Though the majority of the *E. coli* isolates from river water and sediment belong to commensal phylogenetic groups A and B1, rivers are important reservoirs of antibiotic resistance genes where the exchange and transfer of genes can take place among pathogenic and commensal *E. coli* strains, and thus the resistant pathogenic *E. coli* load will increase, and if this results in infection, the disease will be more difficult to treat. There is an association between phylogenetic groups and the host species; human commensal strains belong mostly to group A and B1, and strains isolates from animals fall mostly in group B1. The presence of phylogenetic groups A and B1 can be thus considered an indicator of anthropogenic activities [66]. The reservoir of resistance genes in rivers can thus serve as a potential source of resistance genes to other bacteria as river collects surface water with resistant bacteria from various sources, e.g., municipal wastewater, agricultural activities, etc. Previous studies have reported that high loads of antibiotic resistance gene pollution in the water of rivers resulted from various anthropogenic pollution sources [67,68].

The pollutants of a river deteriorate and alter the water quality, affecting the occurrence and the level of antibiotics in the river water. Further, these pollutants might cause a shift in the bacterial community composition [69,70]. In addition, the bacterial community influences the shaping, abundance, and diversity of antibiotic resistance [71]. As mentioned, Kshipra river receives pollutants from domestic and industrial wastewater, animals and agriculture activities, runoff events, and other inputs such as bathing and washing, especially on festive occasions including Simhastha Kumbh Mela. Regarding Simhastha Kumbh Mela 2016, it might have contributed to the highest presence of *E. coli* resistant to different antibiotics found in the third year. The current results have shown that some of the water quality parameters of Kshipra river exceeded permissible limits including conductivity, total dissolved solids, dissolved oxygen, turbidity, biochemical oxygen demand, chemical oxygen demand, total alkalinity, total hardness, total phosphorus, total coliform, and total *E. coli* (Appendix A). It was also seen that antibiotic residues and antibiotic-resistant *E. coli* in water and sediments were associated with measured water quality parameters. Water quality might contribute to the persistence and the bioavailability of antibiotics in aquatic environments. Antibiotics that occur in the environment in a bioavailable form may therefore pose a risk with respect to antibiotic resistance development [72]. For example, pH, temperature, chloride, and nitrate affect processes such as degradation (photo-degradation by sunlight and biodegradation by bacteria under aerobic and anaerobic conditions) and sorption to organic particles. These processes determine the natural removal, stability, mobility, and transport of antibiotics in the environment [73,74]. In addition, the water quality such as the water temperature and the concentrations of carbon resources, phosphorus, and nitrogen are important factors driving the bacterial community composition, diversity, and dynamics [75]. For instance, temperature and dissolved oxygen have significant associations with the frequency of resistance of *E. coli* to ampicillin; high temperature increases the natural transformation frequencies and rates among the bacterial community in the environment; pH, temperature, and alkaline conditions can affect the hydrolysis and degradation of β-lactams; and total dissolved solid may be involved in the selection process for antibiotic resistance [76,77,78,79]. Previous studies showed antibiotic-resistant *E. coli* and antibiotic resistance genes in aquatic environments as impacted by the bacterial community and water quality parameters in Semenyih river, Malaysia, Ganga river and Cauvery river, India, and East Tiaoxi river and Wen-Rui Tang river, China [71,80,81,82,83]. These studies, however, generally measured individual samples and did not include all the parameters we analysed.

Considering the earlier discussion, it is necessary to include regular monitoring and routine surveillance of water quality, antibiotic residues, and antibiotic resistance of all rivers including Kshipra river in the national and Global Action Plans on containmentof antimicrobial resistance [1,84]. 

## 5. Conclusions

Antibiotic residues and antibiotic-resistant *E. coli* were present in the water and sediment of the Kshipra river during different seasons, sampling sites, and years. Sulfamethoxazole was the most frequently detected antibiotic in the water in different seasons and sites and was significantly associated with the water quality parameters. Other antibiotics such as ofloxacin were also positively correlated with some measured water quality parameters. Significant seasonal and spatial variations of antibiotic-resistant *E. coli* in water and sediment were found. Antibiotic-resistant *E. coli* in the water and sediments was associated with measured water quality parameters. Thus, routine antibiotic residue and antibiotic resistance surveillance and monitoring of the Kshipra river are required to control emerging resistance and its dissemination, which represents a potential threat to human health and the environment. Further, scientifically assessed standards for acceptable levels of antibiotic residues and antibiotic resistance are needed to inform policy and prioritize interventions.

## Figures and Tables

**Figure 1 ijerph-17-07706-f001:**
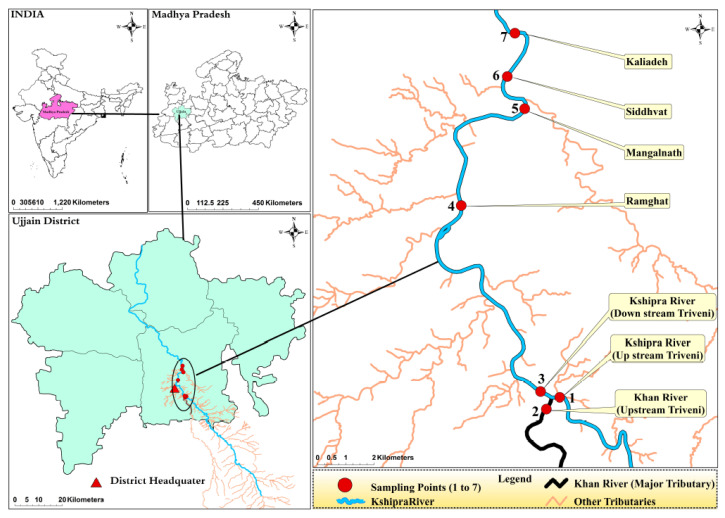
Geographical location of the study site. The map shows (clockwise) India, Madhya Pradesh, the sampling points on the Kshipra river, and Ujjain district. Note: Necessary required data are digitized into shape files. Spatial formatted vector data are converted into shape files and georeferenced. Lambert Conformal Conic Projection is selected to georeference the shapefile. All shapefile features, i.e., the River Kshipra, Khan and minor tributaries, were acquired from base map of ArcInfo, and Google image. Locations of study sampling points were obtained through a field survey using GPS. ArcMap 10.7.1 (ESRI, Redlands, CA, USA) was used to develop all lines as well as for point vectorization and map composition. Map boundaries of India, Madhya Pradesh and districts were taken from the GADM database (https://gadm.org/) where these are available freely for academic purposes. Jammu and Kashmir and Ladakh were treated as one administrative division as the boundary maps separating the two were not available in the retrieved databases.

**Figure 2 ijerph-17-07706-f002:**
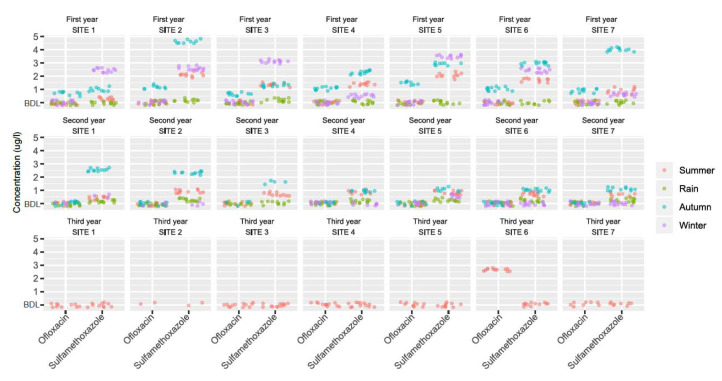
Concentrations of antibiotic residues measured in waters of the Kshipra river in India in various seasons and at various sites over a 3-year period. Note: significant (*p* < 0.05) seasonal and spatial variations in the occurrence of sulfamethoxazole and ofloxacin were found over a 3-year period. Information for all antibiotics is available in Appendix A.

**Figure 3 ijerph-17-07706-f003:**
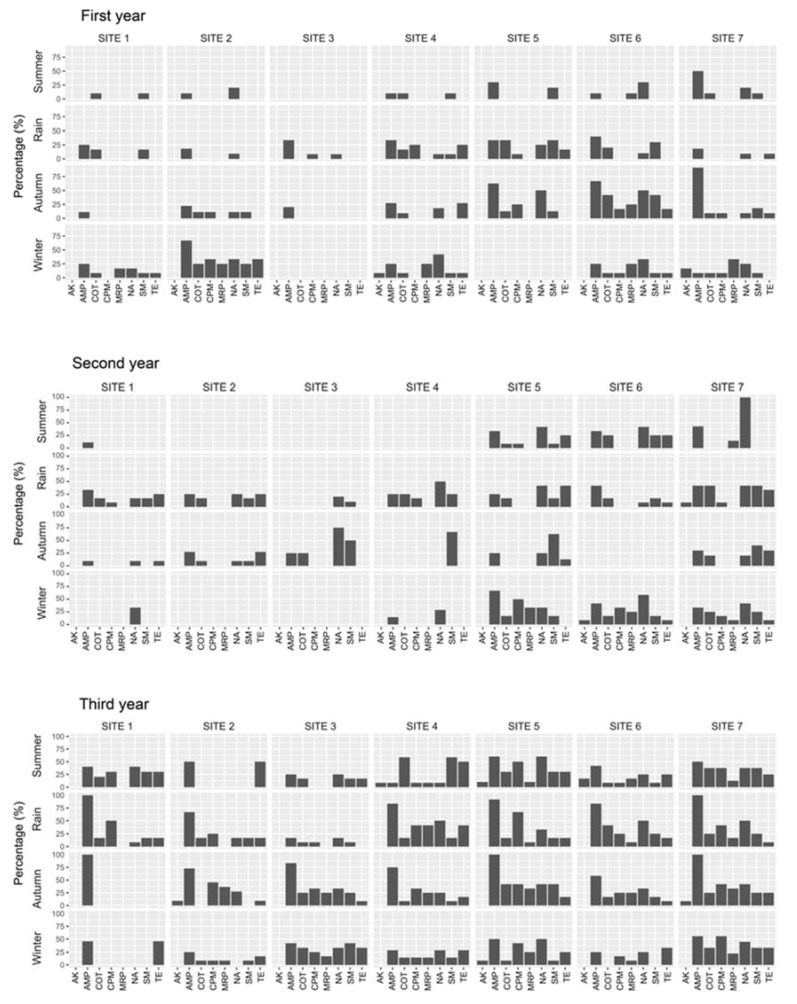
Antibiotic resistance patterns in *E. coli* isolated from water samples of the Kshipra river in India in various seasons and at various sites over a 3-year period. Note: significant (*p* < 0.05) seasonal and spatial variations in the resistance of *E. coli* to amikacin, ampicillin, co-trimoxazole, cefepime, meropenem, nalidixic acid, sulfamethizole, and tetracycline were found over the 3-year period. Information for all antibiotics is available in Table 1. Abbreviations: AK: Amikacin, AMP: Ampicillin, COT: Co-trimoxazole, CPM: Cefepime, MRP: Meropenem, NA: Nalidixic Acid, SM: Sulfamethizole, TE: Tetracycline.

**Figure 4 ijerph-17-07706-f004:**
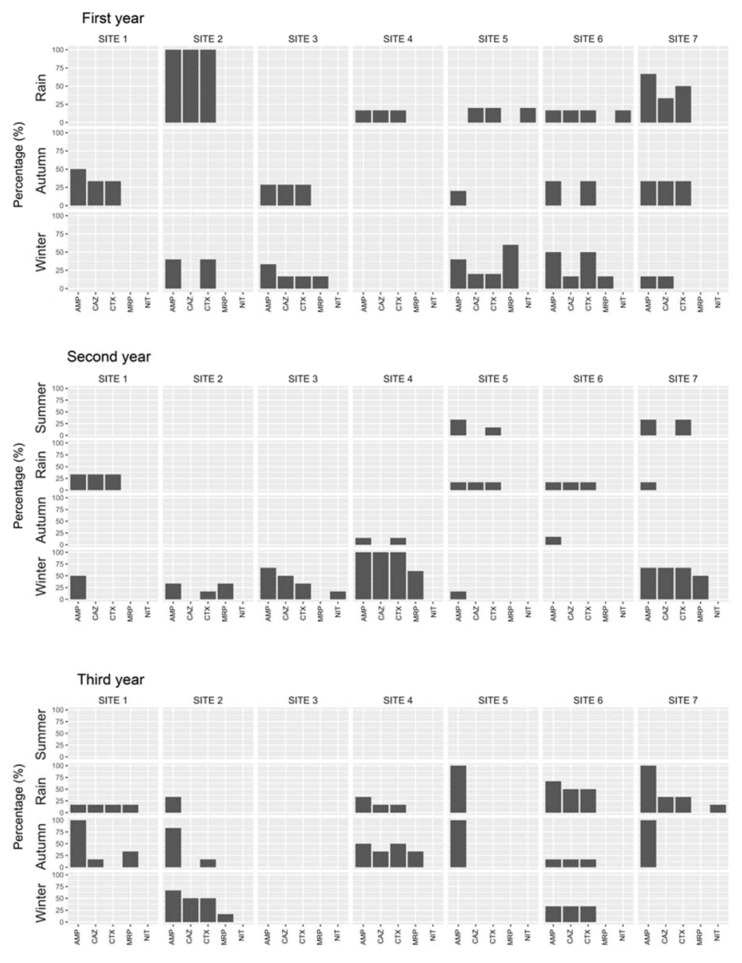
Antibiotic resistance patterns in *E. coli* isolated from sediment samples of the Kshipra river in India in various seasons and at various sites over a 3-year period. Abbreviations: AMP: Ampicillin, CAZ: Ceftazidime, CTX: Cefotaxime, MRP: Meropenem, NIT: Nitrofurantoin. Note: significant (*p* < 0.05) seasonal and spatial variations in the resistance of *E. coli* to ampicillin, ceftazidime, cefotaxime, meropenem, and nitrofurantoin were found over the 3-year period. Information for all antibiotics is available in Appendix A.

**Table ijerph-17-07706-t001a:** (**A**)

Antibiotic	First Year																											
	Summer							Rain							Autumn							Winter						
	Site 1 (N = 10)	Site 2 (N = 10)	Site 3 (N = 10)	Site 4 (N = 10)	Site 5 (N = 10)	Site 6 (N = 10)	Site 7 (N = 10)	Site 1 (N = 12)	Site 2 (N = 11)	Site 3 (N = 12)	Site 4 (N = 12)	Site 5 (N = 12)	Site 6 (N = 10)	Site 7 (N = 11)	Site 1 (N = 9)	Site 2 (N = 9)	Site 3 (N = 10)	Site 4 (N = 11)	Site 5 (N = 8)	Site 6 (N = 12)	Site 7 (N = 11)	Site 1 (N = 12)	Site 2 (N = 12)	Site 3 (N = 12)	Site 4 (N = 12)	Site 5 (N = 11)	Site 6 (N = 12)	Site 7 (N = 12)
	n (%)	n (%)	n (%)	n (%)	n (%)	n (%)	n (%)	n (%)	n (%)	n (%)	n (%)	n (%)	n (%)	n (%)	n (%)	n (%)	n (%)	n (%)	n (%)	n (%)	n (%)	n (%)	n (%)	n (%)	n (%)	n (%)	n (%)	n (%)
Ampicillin	0 (0)	1 (10)	0 (0)	1 (10)	3 (30)	1 (10)	5 (50)	3 (25)	2 (18)	4 (33)	4 (33)	4 (33)	4 (40)	2 (18)	1 (11)	2 (22)	2 (20)	3 (27)	5 (62)	8 (67)	10 (91)	3 (25)	8 (67)	0 (0)	3 (25)	0 (0)	3 (25)	1 (8)
Cefotaxime	2 (20)	1 (10)	0 (0)	0 (0)	2 (20)	0 (0)	4 (40)	0 (0)	1 (9)	2 (17)	3 (25)	4 (33)	2 (20)	1 (9)	0 (0)	1 (11)	2 (20)	1 (9)	4 (50)	2 (17)	4 (36)	2 (17)	5 (42)	0 (0)	1 (8)	0 (0)	3 (25)	1 (8)
Ceftazidime	0 (0)	1 (10)	0 (0)	0 (0)	2 (20)	1 (10)	4 (40)	0 (0)	1 (9)	2 (17)	3 (25)	1 (8)	0 (0)	0 (0)	0 (0)	0 (0)	1 (10)	0 (0)	3 (37)	2 (17)	3 (27)	1 (8)	4 (33)	0 (0)	0 (0)	0 (0)	3 (25)	1 (8)
Cefepime	0 (0)	0 (0)	0 (0)	0 (0)	0 (0)	0 (0)	0 (0)	0 (0)	0 (0)	1 (8)	3 (25)	1 (8)	0 (0)	0 (0)	0 (0)	1 (11)	0 (0)	0 (0)	2 (25)	2 (17)	1 (9)	0 (0)	4 (33)	0 (0)	0 (0)	0 (0)	1 (8)	1 (8)
Amikacin	0 (0)	0 (0)	0 (0)	0 (0)	0 (0)	0 (0)	0 (0)	0 (0)	0 (0)	0 (0)	0 (0)	0 (0)	0 (0)	0 (0)	0 (0)	0 (0)	0 (0)	0 (0)	0 (0)	0 (0)	0 (0)	0 (0)	0 (0)	0 (0)	1 (8)	0 (0)	0 (0)	2 (16)
Gentamicin	0 (0)	0 (0)	0 (0)	0 (0)	0 (0)	0 (0)	0 (0)	0 (0)	0 (0)	0 (0)	0 (0)	0 (0)	0 (0)	0 (0)	0 (0)	0 (0)	0 (0)	0 (0)	0 (0)	0 (0)	0 (0)	0 (0)	0 (0)	0 (0)	0 (0)	0 (0)	0 (0)	0 (0)
Tetracycline	0 (0)	0 (0)	0 (0)	0 (0)	0 (0)	0 (0)	0 (0)	0 (0)	0 (0)	0 (0)	3 (25)	2 (17)	0 (0)	1 (9)	0 (0)	0 (0)	0 (0)	3 (27)	0 (0)	2 (17)	1 (9)	1 (8)	4 (33)	0 (0)	1 (8)	0 (0)	1 (8)	0 (0)
Tigecycline	0 (0)	0 (0)	0 (0)	0 (0)	0 (0)	0 (0)	0 (0)	0 (0)	0 (0)	0 (0)	0 (0)	0 (0)	0 (0)	0 (0)	0 (0)	0 (0)	0 (0)	0 (0)	0 (0)	0 (0)	0 (0)	0 (0)	0 (0)	0 (0)	0 (0)	0 (0)	0 (0)	1 (8)
Imipenem	0 (0)	0 (0)	0 (0)	0 (0)	0 (0)	0 (0)	0 (0)	0 (0)	0 (0)	0 (0)	0 (0)	0 (0)	0 (0)	0 (0)	0 (0)	0 (0)	0 (0)	0 (0)	0 (0)	1 (8)	0 (0)	1 (8)	0 (0)	0 (0)	0 (0)	0 (0)	0 (0)	0 (0)
Meropenem	0 (0)	0 (0)	0 (0)	0 (0)	0 (0)	1 (10)	0 (0)	0 (0)	0 (0)	0 (0)	0 (0)	0 (0)	0 (0)	0 (0)	0 (0)	0 (0)	0 (0)	0 (0)	0 (0)	3 (25)	0 (0)	2 (17)	3 (25)	0 (0)	3 (25)	0 (0)	3 (25)	4 (33)
NalidixicAcid	0 (0)	2 (20)	0 (0)	0 (0)	0 (0)	3 (30)	2 (20)	0 (0)	1 (9)	1 (8)	1 (8)	3 (25)	1 (10)	1 (9)	0 (0)	1 (11)	0 (0)	2 (18)	4 (50)	6 (50)	1 (9)	2 (17)	4 (33)	0 (0)	5 (42)	0 (0)	4 (33)	3 (25)
Ciprofloxacin	0 (0)	0 (0)	0 (0)	0 (0)	0 (0)	1 (10)	0 (0)	0 (0)	0 (0)	0 (0)	3 (25)	2 (17)	1 (10)	0 (0)	0 (0)	0 (0)	0 (0)	0 (0)	3 (38)	3 (25)	2 (18)	1 (8)	3 (25)	0 (0)	1 (8)	0 (0)	2 (17)	1 (8)
Co-trimoxazole	1 (10)	0 (0)	0 (0)	1 (10)	0 (0)	0 (0)	3 (30)	2 (17)	0 (0)	0 (0)	2 (17)	4 (33)	3 (30)	0 (0)	0 (0)	1 (11)	1 (10)	1 (9)	1 (13)	5 (42)	4 (36)	2 (17)	3 (25)	0 (0)	2 (17)	0 (0)	2 (17)	1 (8)
Sulfamethizole	1 (10)	0 (0)	0 (0)	1 (10)	2 (20)	0 (0)	1 (10)	2 (17)	0 (0)	0 (0)	1 (8)	4 (33)	3 (30)	0 (0)	0 (0)	1 (11)	0 (0)	0 (0)	1 (13)	5 (42)	2 (18)	1 (8)	3 (25)	0 (0)	1 (8)	0 (0)	1 (8)	1 (8)
Nitrofurantoin	0 (0)	0 (0)	0 (0)	0 (0)	0 (0)	0 (0)	0 (0)	1 (8)	0 (0)	0 (0)	1 (8)	0 (0)	0 (0)	0 (0)	0 (0)	1 (11)	0 (0)	0 (0)	1 (13)	0 (0)	1 (9)	0 (0)	0 (0)	0 (0)	0 (0)	0 (0)	0 (0)	0 (0)
ESBL	0 (0)	1 (10)	0 (0)	0 (0)	2 (20)	0 (0)	4 (36)	0 (0)	0 (0)	1 (8)	2 (17)	3 (25)	2 (20)	1 (9)	0 (0)	1 (11)	2 (20)	1 (9)	2 (25)	0 (0)	4 (36)	1 (8)	1 (8)	0 (0)	1 (8)	0 (0)	1 (8)	1 (8)
MDR	0 (0)	0 (0)	0 (0)	0 (0)	2 (20)	0 (0)	0 (0)	2 (17)	0 (0)	1 (8)	3 (25)	5 (42)	3 (30)	1 (9)	0 (0)	2 (22)	1 (10)	0 (0)	4 (50)	7 (58)	5 (45)	3 (25)	7 (58)	1 (8)	3 (25)	0 (0)	4 (33)	2 (17)

**Table ijerph-17-07706-t001b:** (**B**)

Antibiotic	Second Year																											
	Summer							Rain							Autumn							Winter						
	Site 1 (N = 9)	Site 2 (N = 9)	Site 3 (N = 9)	Site 4 (N = 9)	Site 5 (N = 9)	Site 6 (N = 9)	Site 7 (N = 9)	Site 1 (N = 9)	Site 2 (N = 9)	Site 3 (N = 9)	Site 4 (N = 9)	Site 5 (N = 9)	Site 6 (N = 9)	Site 7 (N = 9)	Site 1 (N = 9)	Site 2 (N = 9)	Site 3 (N = 9)	Site 4 (N = 9)	Site 5 (N = 9)	Site 6 (N = 9)	Site 7 (N = 9)	Site 1 (N = 9)	Site 2 (N = 9)	Site 3 (N = 9)	Site 4 (N = 9)	Site 5 (N = 9)	Site 6 (N = 9)	Site 7 (N = 9)
	n (%)	n (%)	n (%)	n (%)	n (%)	n (%)	n (%)	n (%)	n (%)	n (%)	n (%)	n (%)	n (%)	n (%)	n (%)	n (%)	n (%)	n (%)	n (%)	n (%)	n (%)	n (%)	n (%)	n (%)	n (%)	n (%)	n (%)	n (%)
Ampicillin	1 (11)	0 (0)	0 (0)	0 (0)	4 (33)	4 (33)	3 (43)	4 (33)	3 (25)	0 (09)	3 (25)	3 (25)	5 (42)	5 (42)	1 (10)	3 (27)	1 (25)	0 (0)	2 (25)	0 (0)	3 (30)	0 (0)	0 (0)	0 (09	1 (15)	4 (67)	5 (42)	4 (33)
Cefotaxime	1 (11)	0 (0)	0 (0)	0 (0)	3 (25)	1 (8)	2 (29)	4 (33)	1 (8)	1 (10)	4 (33)	3 (25)	2 (17)	4 (33)	1 (9)	0 (0)	1 (25)	0 (0)	2 (25)	1 (8)	2 (20)	0 (0)	0 (0)	0 (0)	0 (0)	3 (50)	3 (25)	2 (17)
Ceftazidime	1 (11)	0 (0)	0 (0)	0 (0)	3 (3)	1 (8)	2 (29)	2 (17)	1 (8)	0 (0)	2 (17)	1 (8)	1 (8)	3 (25)	0 (0)	0 (0)	0 (0)	0 (0)	1 (13)	1 (8)	1 (10)	0 (0)	0 (0)	0 (0)	0 (0)	3 (50)	4 (33)	2 (17)
Cefepime	0 (0)	0 (0)	0 (0)	0 (0)	1 (8)	0 (0)	0 (0)	1 (8)	0 (0)	0 (0)	2 (17)	0 (0)	0 (0)	1 (8)	0 (0)	0 (0)	0 (0)	0 (0)	0 (0)	0 (0)	0 (0)	0 (0)	0 (0)	0 (0)	0 (0)	3 (50)	4 (33)	2 (17)
Amikacin	0 (0)	0 (0)	0 (0)	0 (0)	0 (0)	0 (0)	0 (0)	0 (0)	0 (0)	0 (0)	0 (0)	0 (0)	0 (0)	1 (8)	0 (0)	0 (0)	0 (0)	0 (0)	0 (0)	0 (0)	0 (0)	0 (0)	0 (0)	0 (0)	0 (0)	0 (0)	1 (8)	0 (0)
Gentamicin	0 (0)	0 (0)	0 (0)	0 (0)	0 (0)	0 (0)	0 (0)	0 (0)	0 (0)	0 (0)	1 (8)	0 (0)	0 (0)	0 (0)	0 (0)	0 (0)	0 (0)	0 (0)	0 (0)	0 (0)	0 (0)	0 (0)	0 (0)	0 (0)	0 (0)	0 (0)	0 (0)	0 (0)
Tetracycline	0 (0)	0 (0)	0 (0)	0 (0)	3 (25)	3 (25)	0 (0)	3 (25)	3 (25)	0 (0)	0 (0)	5 (42)	1 (8)	4 (33)	1 (9)	3 (27)	0 (0)	0 (0)	1 (12)	0 (0)	3 (30)	0 (0)	0 (0)	0 (0)	0 (0)	0 (0)	1 (8)	1 (8)
Tigecycline	0 (0)	0 (0)	0 (0)	0 (0)	0 (0)	0 (0)	0 (0)	0 (0)	0 (0)	0 (0)	0 (0)	0 (0)	0 (0)	0 (0)	0 (0)	0 (0)	0 (0)	0 (0)	0 (0)	0 (0)	0 (0)	0 (0)	0 (0)	0 (0)	0 (0)	0 (0)	0 (0)	0 (0)
Imipenem	0 (0)	0 (0)	0 (0)	0 (0)	0 (0)	0 (0)	1 (14)	0 (0)	0 (0)	0 (0)	1 (8)	1 (8)	0 (0)	0 (0)	1 (9)	0 (0)	0 (0)	2 (22)	0 (0)	0 (0)	0 (0)	0 (0)	0 (0)	0 (0)	0 (0)	0 (0)	2 (17)	0 (0)
Meropenem	0 (0)	0 (0)	0 (0)	0 (0)	0 (0)	0 (0)	1 (14)	0 (0)	0 (0)	0 (0)	0 (0)	0 (0)	0 (0)	0 (0)	0 (0)	0 (0)	0 (0)	0 (0)	0 (0)	0 (0)	0 (0)	0 (0)	0 (0)	0 (0)	0 (0)	2 (33)	3 (25)	1 (8)
NalidixicAcid	0 (0)	0 (0)	0 (0)	0 (0)	5 (42)	5 (42)	7 (100)	2 (17)	3 (25)	2 (20)	6 (50)	5 (42)	1 (8)	5 (42)	1 (9)	1 (9)	3 (75)	0 (0)	2 (25)	0 (0)	2 (20)	1 (33)	0 (0)	0 (0)	2 (29)	2 (33)	7 (58)	5 (42)
Ciprofloxacin	0 (0)	0 (0)	0 (0)	0 (0)	3 (25)	5 (42)	3 (43)	1 (8)	3 (25)	1 (10)	3 (25)	3 (25)	1 (8)	1 (8)	1 (9)	0 (0)	1 (25)	0 (0)	1 (13)	0 (0)	2 (20)	0 (0)	0 (0)	0 (0)	2 (29)	2 (33)	3 (25)	1 (8)
Co-trimoxazole	0 (0)	0 (0)	0 (0)	0 (0)	2 (17)	3 (25)	0 (0)	2 (17)	2 (17)	1 (10)	3 (25)	2 (17)	2 (17)	6 (50)	0 (0)	1 (9)	1 (25)	0 (0)	0 (0)	0 (0)	2 (20)	0 (0)	0 (0)	0 (0)	0 (0)	1 (17)	2 (17)	3 (25)
Sulfamethizole	0 (0)	0 (0)	0 (0)	0 (0)	1 (8)	3 (25)	0 (0)	2 (17)	2 (17)	1 (10)	3 (25)	2 (17)	2 (17)	5 (42)	0 (0)	1 (9)	2 (50)	6 (67)	5 (63)	0 (0)	4 (40)	0 (0)	0 (0)	0 (0)	0 (0)	1 (17)	2 (17)	3 (25)
Nitrofurantoin	0 (0)	0 (0)	0 (0)	0 (0)	0 (0)	0 (0)	0 (0)	0 (0)	0 (0)	0 (0)	0 (0)	1 (8)	0 (0)	0 (0)	0 (0)	0 (0)	0 (0)	0 (0)	0 (0)	0 (0)	0 (0)	0 (0)	0 (0)	0 (0)	0 (0)	0 (0)	0 (0)	0 (0)
ESBL	1 (11)	0 (0)	0 (0)	0 (0)	3 (25)	1 (8)	0 (0)	4 (33)	1 (8)	0 (0)	2 (17)	2 (17)	2 (17)	4 (33)	1 (9)	0 (0)	1 (25)	0 (0)	2 (25)	1 (8)	2 (20)	2 (67)	1 (25)	0 (0)	3 (43)	3 (50)	2 (17)	1 (8)
MDR	1 (11)	0 (0)	0 (0)	0 (0)	4 (33)	3 (25)	2 (29)	3 (25)	3 (25)	2 (25)	3 (25)	4 (33)	2 (17)	6 (50)	1 (10)	3 (27)	2 (50)	5 (56)	1 (13)	0 (0)	3 (30)	0 (0)	0 (0)	0 (0)	0 (0)	2 (33)	5 (42)	3 (25)

**Table ijerph-17-07706-t001c:** (**C**)

Antibiotic	Third Year																											
	Summer							Rain							Autumn							Winter						
	Site 1 (N = 10)	Site 2 (N = 2)	Site 3 (N = 12)	Site 4 (N = 12)	Site 5 (N = 10)	Site 6 (N = 12)	Site 7 (N = 8)	Site 1 (N = 12)	Site 2 (N = 12)	Site 3 (N = 12)	Site 4 (N = 12)	Site 5 (N = 12)	Site 6 (N = 12)	Site 7 (N = 12)	Site 1 (N = 6)	Site 2 (N = 11)	Site 3 (N = 12)	Site 4 (N = 12)	Site 5 (N = 12)	Site 6 (N = 12)	Site 7 (N = 12)	Site 1 (N = 11)	Site 2 (N = 12)	Site 3 (N = 12)	Site 4 (N = 7)	Site 5 (N = 12)	Site 6 (N = 12)	Site 7 (N = 9)
	n (%)	n (%)	n (%)	n (%)	n (%)	n (%)	n (%)	n (%)	n (%)	n (%)	n (%)	n (%)	n (%)	n (%)	n (%)	n (%)	n (%)	n (%)	n (%)	n (%)	n (%)	n (%)	n (%)	n (%)	n (%)	n (%)	n (%)	n (%)
Ampicillin	4 (40)	1 (50)	3 (25)	1 (8)	6 (60)	5 (42)	4 (50)	12 (100)	8 (67)	2 (17)	10 (84)	11 (92)	10 (83)	12 (100)	6 (100)	8 (72)	10 (83)	9 (75)	12 (100)	7 (58)	12 (100)	5 (45)	3 (25)	5 (42)	2 (29)	6 (50)	3 (25)	5 (56)
Cefotaxime	3 (30)	0 (0)	1 (8)	1 (8)	5 (50)	1 (8)	3 (38)	0 (0)	3 (25)	2 (17)	5 (42)	2 (17)	3 (25)	5 (42)	0 (0)	5 (45)	6 (50)	4 (33)	6 (50)	3 (25)	6 (50)	0 (0)	2 (17)	3 (25)	1 (14)	5 (42)	2 (17)	5 (56)
Ceftazidime	3 (30)	0 (0)	1 (8)	1 (89	5 (50)	1 (8)	1 (13)	0 (0)	3 (25)	2 (17)	5 (42)	2 (17)	3 (25)	5 (42)	0 (0)	5 (45)	3 (25)	4 (33)	5 (42)	3 (25)	4 (33)	0 (0)	1 (8)	2 (17)	1 (14)	5 (42)	2 (17)	5 (56)
Cefepime	3 (30)	0 (0)	0 (0)	1 (8)	5 (50)	1 (8)	3 (38)	6 (50)	3 (25)	1 (8)	5 (42)	8 (67)	3 (25)	5 (42)	0 (0)	5 (45)	4 (33)	4 (33)	5 (42)	3 (25)	5 (42)	0 (0)	1 (8)	3 (25)	1 (14)	5 (42)	2 (17)	5 (56)
Amikacin	0 (0)	0 (0)	0 (0)	1 (8)	1 (10)	2 (17)	0 (0)	0 (0)	0 (0)	0 (0)	0 (0)	0 (0)	0 (0)	0 (0)	0 (0)	1 (9)	0 (0)	0 (0)	0 (0)	0 (0)	1 (8)	0 (0)	0 (0)	0 (0)	0 (0)	1 (8)	0 (0)	0 (0)
Gentamicin	0 (0)	0 (0)	0 (0)	1 (8)	1 (10)	2 (17)	0 (0)	0 (0)	0 (0)	0 (0)	0 (0)	1 (8)	0 (0)	0 (0)	0 (0)	1 (9)	0 (0)	0 (0)	0 (0)	0 (0)	1 (8)	0 (0)	0 (0)	0 (0)	0 (0)	2 (17)	0 (0)	0 (0)
Tetracycline	3 (30)	1 (50)	2 (17)	6 (50)	3 (30)	3 (25)	2 (25)	2 (17)	2 (17)	0 (0)	5 (42)	2 (17)	2 (17)	1 (8)	0 (0)	1 (9)	1 (8)	2 (17)	2 (17)	1 (8)	3 (25)	5 (45)	2 (17)	4 (33)	2 (29)	3 (25)	4 (33)	3 (33)
Tigecycline	0 (0)	0 (0)	0 (0)	0 (0)	0 (0)	0 (0)	0 (0)	0 (0)	0 (0)	0 (0)	0 (0)	0 (0)	0 (0)	0 (0)	0 (0)	0 (0)	0 (0)	0 (0)	0 (0)	0 (0)	0 (0)	0 (0)	0 (0)	0 (0)	0 (0)	0 (0)	0 (0)	0 (0)
Imipenem	0 (0)	0 (0)	0 (0)	1 (8)	1 (10)	2 (17)	0 (0)	1 (8)	0 (0)	0 (0)	0 (0)	3 (25)	0 (0)	0 (0)	0 (0)	1 (9)	0 (0)	1 (8)	0 (0)	0 (0)	1 (8)	0 (0)	0 (0)	0 (0)	0 (0)	1 (8)	0 (0)	0 (0)
Meropenem	0 (0)	0 (0)	0 (0)	1 (8)	1 (10)	2 (17)	1 (12)	0 (0)	0 (0)	0 (0)	5 (42)	1 (8)	1 (8)	2 (17)	0 (0)	4 (36)	3 (25)	3 (25)	4 (33)	3 (25)	4 (33)	0 (0)	1 (8)	2 (17)	1 (14)	3 (25)	1 (8)	2 (22)
Nalidixic Acid	4 (40)	0 (0)	3 (25)	1 (8)	6 (60)	3 (25)	3 (38)	1 (8)	2 (17)	2 (17)	6 (50)	4 (33)	6 (50)	6 (50)	0 (0)	3 (27)	4 (33)	3 (25)	5 (42)	4 (33)	5 (42)	0 (0)	0 (0)	4 (33)	2 (29)	6 (50)	3 (25)	4 (44)
Ciprofloxacin	2 (20)	0 (0)	0 (0)	1 (8)	3 (30)	1 (8)	3 (38)	0 (0)	1 (8)	0 (0)	5 (42)	3 (25)	3 (25)	4 (33)	0 (0)	3 (27)	3 (25)	3 (25)	2 (17)	2 (17)	4 (33)	0 (0)	1 (8)	4 (33)	1 (14)	3 (25)	1 (8)	2 (22)
Co-trimoxazole	2 (20)	0 (0)	2 (17)	7 (58)	4 (40)	3 (25)	3 (38)	2 (17)	3 (25)	1 (8)	2 (17)	2 (17)	6 (50)	3 (25)	0 (0)	1 (9)	3 (25)	1 (8)	5 (42)	2 (17)	4 (33)	0 (0)	1 (8)	4 (33)	1 (14)	2 (17)	0 (0)	3 (33)
Sulfamethizole	3 (30)	0 (0)	2 (17)	7 (58)	3 (30)	1 (8)	3 (38)	2 (17)	2 (17)	1 (8)	2 (17)	2 (17)	3 (25)	3 (25)	0 (0)	0 (0)	3 (25)	1 (8)	5 (42)	2 (17)	3 (25)	0 (0)	1 (8)	5 (42)	1 (14)	1 (8)	0 (0)	3 (33)
Nitrofurantoin	1 (10)	1 (50)	0 (0)	0 (0)	0 (0)	0 (0)	0 (0)	0 (0)	1 (8)	1 (8)	1 (8)	0 (0)	0 (0)	0 (0)	0 (0)	1 (9)	0 (0)	0 (0)	1 (8)	1 (8)	0 (0)	0 (0)	0 (0)	0 (0)	0 (0)	0 (0)	0 (0)	0 (0)
ESBL	3 (30)	0 (0)	1 (8)	0 (0)	4 (40)	0 (0)	3 (38)	0 (0)	2 (17)	1 (8)	2 (17)	2 (17)	0 (0)	3 (25)	0 (0)	3 (27)	2 (17)	3 (25)	3 (25)	3 (25)	3 (25)	0 (0)	2 (17)	0 (0)	1 (14)	3 (25)	1 (8)	5 (56)
MDR	5 (50)	1 (50)	3 (25)	7 (58)	4 (40)	5 (42)	4 (50)	5 (42)	4 (33)	2 (17)	6 (50)	6 (50)	8 (67)	8 (67)	0 (0)	4 (36)	6 (50)	4 (33)	8 (67)	5 (42)	5 (42)	4 (36)	2 (17)	5 (42)	1 (14)	5 (42)	3 (25)	5 (56)

**Table ijerph-17-07706-t001d:** (**D**)

Antibiotic	*p*-Value *
Ampicillin	<0.0001
Cefotaxime	0.22
Ceftazidime	0.2
Cefepime	0.0003
Amikacin	0.04
Gentamicin	0.7
Tetracycline	0.0002
Tigecycline	-
Imipenem	0.4
Meropenem	<0.0001
Nalidixic Acid	0.04
Ciprofloxacin	0.11
Co-trimoxazole	0.0459
Sulfamethizole	0.02
Nitrofurantoin	0.63
ESBL	0.06
MDR	0.003

Abbreviations: ESBL: Extended Spectrum Beta-Lactamase, MDR: multidrug resistance, N = total number of *E. coli* isolates, n = number of resistant *E coli* isolates. * *p*-value are extracted from Analysis of Variance (ANOVA), significant seasonal and spatial variations in the resistance of *E. coli* to antibiotics over the 3-year period were determined by *p*-values < 0.05 in Table 1.

**Table 2 ijerph-17-07706-t002:** Antibiotic resistance genes detected in resistant *E. coli* isolated from water samples of the Kshipra river in India in various seasons over a 3-year period.

River Water Samples														
Antibiotic Resistance Genes		First Year				Second Year				Third Year				*p*-Value *
		Summer	Rainy	Autumn	Winter	Summer	Rainy	Autumn	Winter	Summer	Rainy	Autumn	Winter	
		Resistance n (%)	Resistance n (%)	Resistance n (%)	Resistance n (%)	Resistance n (%)	Resistance n (%)	Resistance n (%)	Resistance n (%)	Resistance n (%)	Resistance n (%)	Resistance n (%)	Resistance n (%)	
Cephalosporin resistance genes	*CTX-M1 ^a^*	8 (22)	10 (31)	9 (50)	8 (19)	9 (30)	12 (40)	11 (57)	9 (27)	9 (24)	11 (34)	14 (66)	12 (21)	0.00
	*CTX-M9 ^a^*	0	0	0	0	0	1 (3)	0	0	1 (3)	0	0	0	0.84
Quinolone resistance genes	*Qnr B ^b^*	1 (5)	2 (7)	0	0	0	0	0	0	1 (3)	0	0	0	0.77
	*Qnr S ^b^*	2 (10)	6 (23)	2 (8)	7 (16)	6 (15)	4 (17)	3 (14)	7 (17)	6 (18)	8 (23)	2 (7)	5 (10)	0.27
Sulfa resistance genes	*Sul-I ^c^*	2 (5)	4 (12)	3 (12)	4 (12)	8 (25)	3 (10)	5 (22)	5 (17)	10 (26)	5 (17)	3 (13)	7 (16)	0.82
	*Sul-II ^c^*	4 (10)	6 (18)	5 (20)	4 (12)	5 (15)	9 (31)	5 (22)	5 (17)	4 (10)	7 (25)	7 (31)	7 (16)	0.003

River water samples—first, second, and third year, respectively. ^a^ Summer-35/30/37, Rain-32/30/32, Autumn-18/19/21, Winter-42/33/55. ^b^ Summer-20/40/32, Rain-26/23/34, Autumn-25/21/26, Winter-43/39/49. ^c^ Summer-40/32/38, Rain-33/29/28, Autumn-24/22/22, Winter-33/28/44. * The *p*-value is a comparison of the prevalence of genes in isolates in different seasons over three years.

**Table 3 ijerph-17-07706-t003:** Antibiotic resistance genes detected in resistant *E. coli* isolated from sediment samples of the Kshipra river in India in various seasons over a 3-year period.

River Sediment Samples														
Antibiotic Resistance Genes		First Year				Second Year				Third Year				*p*-Value *
		Summer	Rainy	Autumn	Winter	Summer	Rainy	Autumn	Winter	Summer	Rainy	Autumn	Winter	
		Resistance n (%)	Resistance n (%)	Resistance n (%)	Resistance n (%)	Resistance n (%)	Resistance n (%)	Resistance n (%)	Resistance n (%)	Resistance n (%)	Resistance n (%)	Resistance n (%)	Resistance n (%)	
Cephalosporin resistance genes	*CTX-M1 ^a^*	5 (31)	5 (25)	2 (40)	7 (41)	4 (28)	4 (50)	7 (46)	3 (20)	8 (47)	6 (60)	7 (43)	3 (13)	0.003
	*CTX-M9 ^a^*	0	0	0	0	0	0	0	0	0	0	0	0	-
Quinolone resistance genes	*Qnr B ^b^*	0	0	0	0	0	0	0	0	0	0	0	0	-
	*Qnr S ^b^*	2 (50)	0	3 (23)	2 (22)	2 (16)	3 (25)	2 (50)	2 (100)	2 (28)	2 (20)	2 (40)	1 (25)	0.75
Sulfa resistance genes	*Sul-I ^c^*	2 (28)	1 (25)	3 (33)	2 (33)	3 (33)	2 (66)	1 (14)	2 (66)	2 (33)	1 (20)	4 (80)	0	0.966
	*Sul-II ^c^*	2 (28)	1 (25)	3 (33)	4 (66)	2 (22)	0	2 (28)	0	5 (83)	4 (80)	4 (80)	1 (25)	0.995

River sediment samples—first, second and third year, respectively. ^a^ Summer-16/14/17, Rain-20/8/10, Autumn-5/15/16, Winter-17/15/23. ^b^ Summer-4/12/7, Rain-3/12/10, Autumn-13/4/5, Winter-9/2/4. ^c^ Summer-7/9/6, Rain-4/3/5, Autumn-9/7/5, Winter-6/3/4. * The *p*-value is a comparison of the prevalence of genes in isolates in different seasons over the three years.

**Table 4 ijerph-17-07706-t004:** Phylogenetic grouping of resistant *E. coli* isolated from water and sediment samples of the Kshipra river in India.

Type of Sample	Phylogenetic Groups (n)				*p*-Value
	A	B1	B2	D	
River water (n = 668)	410	154	25	73	0.00641
River sediment (n = 205)	144	30	3	32	

**Table 5 ijerph-17-07706-t005:** Average values (standard deviation) of water quality parameters of the Kshipra river in India in various seasons and at various sites over a 3-year period.

Water Quality Parameter	Mean	SD
Ambient temperature (°C)	28.4	6.2
Water temperature (°C)	25.9	4.9
pH	8.2	0.4
Conductivity (µS/cm)	1035	360.3
Total dissolved solids (mg/L)	652	227.3
Dissolved oxygen (mg/L)	5.9	2.9
Turbidity (NTU)	90.4	157.3
Total suspended solids (mg/L)	124	161.3
Biochemical oxygen demand (mg/L)	26.5	19.1
Chemical oxygen demand (mg/L)	70.9	50.2
Free carbon dioxide (mg/L)	6	7.1
Phenolphthalein alkalinity (mg/L)	4.1	6
Methyl orange alkalinity (mg/L)	317.3	108.5
Total alkalinity (mg/L)	322.8	109.3
Chloride (mg/L)	159.8	67.8
Total hardness (mg/L)	307.6	114.6
Calcium hardness (mg/L)	154.1	63.4
Magnesium hardness (mg/L)	153.4	68.9
Nitrate nitrogen (mg/L)	7.4	3.5
Total phosphorus (mg/L)	3.8	2.2
Ortho phosphorus (mg/L)	3	2.2
Organic phosphorus (mg/L)	0.8	0.9
Total coliform (CFU/100)	76.5	43.7
Total *E. Coli.* (CFU/100)	52.6	31.9

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
