# Peer review of "Monitoring of Water Quality, Antibiotic Residues, and Antibiotic-Resistant Escherichia coli in the Kshipra River in India over a 3-Year Period"

_ijerph, 2020, doi:10.3390/ijerph17217706_

Round 1

Reviewer 1 Report

1.- Explain better what is meant by sampling with contaminated and uncontaminated points.

2.- Please indicated the reference of the quantity of water sampled. 

Reviewer 2 Report

This paper looks the quality of water, the presence of antibiotic residues and the antibiotic resistance patterns (and presence of AMR genes) in E.coli from the Kshipra river in India over a 3-year period.

The paper is very well written and contributes greatly to the information concerning the spread of antibiotic resistance in this part of the world.

I recommend acceptance subject to some minor changes.

  1. In "2.5. Water and sediment quality parameters examined in the laboratory" the authors mention tests that were carried and give two references. Please expand the information given here. What actual test were done?
  2. In "Microbiological methods" the authors say tests were carried out "according to standard methods" please expand on this.
  3. What controls were used when testing the E. coli isolate antibiotic resistance pattern
  4. Presumably results were compared to Enterobacterales breakpoints?
  5. Please include the PCR primer and methods used in"Molecular methods" section (in supplementary data would be fine)
  6. Figure legends should be at the bottom of Figures.
  7. Some of the figures are difficult to interpret due to both the level of information contained an the fact that the text is fuzzy.
  8. In line 343 the authors say "Though the majority of the E. coli isolates from river water and sediment belonged to commensal phylogenetic groups A and B1" Please add a brief explanation explaining why this matters

Reviewer 3 Report

The manuscript presents the outcome of a study of the water quality of several parameters over the course of three years.

Although the article is very interesting and with good results I find it unbalanced to the antibiotic residues part with missing details in the microbial experiments and results.

In detail:

Page2, line51:” Various factors affect the concentrations of detected antibiotics in the aquatic environment, such as the use of antibiotics in different periods over time and environmental behaviors of behaviors of antibiotics such as degradation, sorption and water quality”. Can the authors explain the water quality part at the end of the sentence since it does not make sense

Page 2, Line 54, 55: “…such as discharge of domestic sewage, industrial effluent [9]–[11], hospital and municipal wastewater”. Municipal wastewater and domestic sewage are the same… correct?

Page 4, Lines 132, 133: Protocol needs to be described briefly.

Page 4, secton 2.5: What were the method used for all this determinations?

Page 5, section 2.7. Methods need to be explained briefly

Page 5, Section 2.8. Again the methods need to be explained and the list of primers/probes used needs to be presented

Page 6, Figure 2: This figure is difficult to red, it must be changed for a better perception of the results

Figures 3 and 4: What do the y-axis represents? Percentage? This must be explicit in the graphics

Table S5: Why all the results dealing with e. coli were passed to supplementary information? It is important and relevant to the article that they are included in the manuscript. In addition, how many e. coli isolates were tested in total? Only 10 in the total of 3 years?

Table s11: The results of the chemical parameters need to be included in the paper for better understanding of the results. The authors shouldn’t present correlations between them

Round 2

Reviewer 3 Report

The authors corrected and improved the manuscript. The manuscript can be accepted as is.